# The Prevalence of Vitamin and Mineral Deficiencies and High Levels of Non-Essential Heavy Metals in Saudi Arabian Adults

**DOI:** 10.3390/healthcare10122415

**Published:** 2022-11-30

**Authors:** Omar Albalawi, Rasha Almubark, Abdulrahman Almarshad, Amani S. Alqahtani

**Affiliations:** 1Research & Studies, Saudi Food and Drug Authority, Riyadh 13513, Saudi Arabia; 2Project Management Office, Saudi Food and Drug Authority, Riyadh 13513, Saudi Arabia

**Keywords:** micronutrient, vitamins, minerals, metals, prevalence, national health policy, epidemiological, deficiencies, food fortification

## Abstract

Although the micronutrient status of a population is of high national priority, as it critically impacts public health, limited data is available for quantifying the micronutrient status in Saudi Arabia. We aimed to provide comprehensive, epidemiological, descriptive data regarding micronutrient levels in Saudi adults. This cross-sectional study included 3432 adults aged ≥18 years (mean age, 37.7 ± 11.7 years; women, 51.2%) across all 13 Saudi administrative regions (March 2019–November 2021). Laboratory data for 14 micronutrients (8 vitamins, 4 minerals, and 2 nonessential heavy metals) were characterized using descriptive analysis. Vitamin D deficiency (64.3%) was the most prevalent, followed by vitamin B2 (44.9%) and A (9.6%) deficiencies. Among minerals, iron deficiency was the most prevalent (23.2%), followed by zinc (15.3%) and copper (8.7%) deficiencies. Most Saudi adults exhibited normal arsenic (99.7%) and mercury (99.9%) levels. Men exhibited significantly higher vitamin B2, B9, and D deficiencies than women, while women exhibited higher vitamin A, B12, iron, and zinc deficiencies than men. Younger adults demonstrated a significantly higher prevalence of vitamin D and iron deficiencies, whereas older adults exhibited a higher prevalence of vitamin B1 and magnesium deficiencies. As micronutrient deficiencies are a public health concern, health policies and programs need to be developed and implemented to address them.

## 1. Introduction

Micronutrients comprise a major nutrient group, representing vitamins and minerals [1]. They act as cofactors, optimizing health and preventing or treating diseases [2]. The nutritional status of a micronutrient is determined using a continuum scale ranging from deficiency to excess [3,4,5]. Micronutrient deficiencies, also called hidden hunger, are more common than their excess (i.e., toxicities), approximately affecting two billion people worldwide [4]. The most common micronutrient deficiencies include those of iron, iodine, folate (vitamin B9), vitamin A, zinc, and, to a lesser extent, vitamin B12 and other B vitamins [6]. Poor dietary habits constitute the primary factor that can precipitate micronutrient deficiencies [4]. Micronutrient deficiencies are associated with several health conditions and can be detrimental to health if left untreated [1,4].

Furthermore, alongside micronutrient deficiencies, heavy metal exposure in high doses might have harmful effects on humans, which is a major global health concern [7]. Arsenic and mercury are well-known toxic nonessential heavy metals, exhibiting either acute or chronic toxicity [8]. Arsenic-contaminated groundwater is found worldwide, with the highest prevalence in Asia [9,10]. In Saudi Arabia, the consumption of rice irrigated with arsenic-contaminated groundwater [9] constitutes the primary risk factor for arsenic exposure [11]. Mercury is one of the top 10 chemicals associated with significant public health concerns [12]. Fish consumption is a critical source of human exposure to mercury alongside dental amalgam use or occupational exposure [12,13]. A previous study reported that consuming fish from the Red Sea and Arabian Gulf potentially poses a health risk because of mercury bioaccumulation [14]

Research regarding the prevalence of micronutrient deficiencies in Saudi adults appears limited, mainly focusing on vitamin D [15]. To the best of our knowledge, only one study has been conducted at the national level in Saudi Arabia, which used a cutoff of ≤28 ng/mL for vitamin D (25[O.H.]D) deficiency and found an overall vitamin D deficiency among 62.65% and 40.6% of females and males aged 15 years and older, respectively [16]. Thus, national studies examining micronutrient status and heavy metal levels among the Saudi population are sparse.

This study aimed to provide a comprehensive, epidemiological, descriptive overview of the micronutrient status of Saudi adults, examining several vitamins (A, B1, B2, B6, B9, B12, D, and E) and minerals (copper, zinc, magnesium, and iron). Furthermore, we assessed the levels of two well-known toxic heavy metals, arsenic, and mercury, in the Saudi Arabian population. These insights could provide holistic evidence for the present burden of vitamin and mineral deficiencies and heavy metal levels among Saudi adults and aid policymakers in developing early interventions to tackle these issues.

## 2. Materials and Methods

### 2.1. Study Design

This nationwide cross-sectional study included Saudi adults (≥18 years of age) living across the 13 administrative regions of Saudi Arabia (Riyadh, Makkah Al Mukaramah, Eastern Region, Northern Borders, Madinah, Jezan, Asir, Najran, Qassim, Tabuk, Hail, Al-Jouf, and Al-Baha) and was conducted between September 2019 and June 2021, with interruptions in data collection from March to November 2020 because of the COVID-19 pandemic.

### 2.2. Sampling and Sample Size

The quota sampling technique was used to achieve an equal distribution of respondents. The participants were stratified by age based on the Saudi Arabia median age of 37 years [17]. Following additional stratification by sex (men and women), a total of 52 quotas were created for this study (i.e., 4 quotas in each administrative region). The sample size was determined based on a medium effect size of ~0.3, with 80% power and a 95% confidence interval. Thus, each quota comprised a sample size of 68 participants, and the total targeted sample was 3536 participants.

### 2.3. Participants and Recruitment Method

The following eligibility criteria were applied: age (≥18 years of age), residency status (currently living in Saudi Arabia), and citizenship status (i.e., Saudi Arabian citizen).

A list of random phone numbers based on age and sex information derived from different governmental databases was generated to ensure sample size diversity. A trained data collection team recruited the participants via a phone call survey using a computer-assisted telephone interview to determine the respondents who fit in the stratum quota, who were then invited to complete the study.

The eligible participants were recruited through two phases: phase I (survey) and phase II (laboratory measurements).

#### 2.3.1. Phase I (Survey)

A research team comprising a nutritionist, pharmacist, and public health expert developed the questionnaire used in this study. The questionnaire was then reviewed and revised by other experts (a clinical laboratory technician and a medical doctor). Linguistic validation was performed with a focus group comprising 15 participants, who were selected from different backgrounds to ensure the clarity, intelligibility, and appropriateness as well as the cultural relevance of the study questions. In addition, a pilot study was conducted among 100 participants to test the entire study protocol and questionnaire design, which were updated accordingly.

Once selected, the individuals were contacted, and verbal consent was obtained. The participants were asked to provide their demographic information (nationality, age, sex, and region) to confirm the participant met the eligibility requirements. Subsequently, the interviewer delivered a structured questionnaire including questions regarding tobacco use status, diagnosed chronic conditions, and dietary supplement and medication use. Respondents were considered tobacco users if they participated in any form of nicotine smoking (cigarettes, water pipes, or e-cigarettes) and were classified as current smokers, former smokers, or never smoked.

For participants with chronic diseases lasting 1 year or more and requiring ongoing medical attention, morbidity was classified based on the four primary chronic disease categories defined by the World Health Organization (WHO): cardiovascular diseases (such as heart disease or stroke), cancers, chronic respiratory diseases, and diabetes [18]. In addition, we considered chronic mental health conditions (depression), similar to previous national studies [19]. Hence, health conditions were classified into three mutually exclusive groups (no condition, one condition, and more than two conditions).

#### 2.3.2. Phase II (Laboratory Measurements)

Within 3 days of completing the survey, a blood test was performed for each participant, for which they were referred to contracted medical laboratories across the nation. For this phase, a written consent form was obtained from each participant. Blood samples were collected from the median cubital and cephalic veins and tested for serum vitamin (A, B1, B2, B6, B9, D, and E), mineral (copper, ferritin, manganese, and zinc), and heavy metal (arsenic and mercury) levels. The results of each test item for vitamins and minerals were categorized as deficient or not deficient based on the reference range of each item. The laboratory findings of heavy metal levels were categorized as normal or level (Appendix A
Table A1).

### 2.4. Data Analysis

All continuous biomarkers presented here were dichotomized as deficient or not deficient based on the baseline reference range for descriptive purposes (Appendix A
Table A1). Chi-squared tests were used to compare the prevalence of micronutrient outcomes between sex and age groups. The calculated *p*-values were two-tailed, and *p* < 0.05 was considered significant. Fisher’s exact tests were used for any expected values less than one. A poststratification adjustment was performed based on 2018 Saudi Arabia census data defined by administrative regions, ensuring that the joint distribution of regions in the weighted sample matched the known Saudi Arabian population’s joint distribution [17]. Data analyses were performed using the Statistical Package for Social Sciences (SPSS, Armonk, NY, USA).

### 2.5. Ethical Considerations

This study was approved by the Institutional Review Board of the Saudi Food & Drug Authority (SFDA) (approval no. 0008-19). Verbal consent to participate was obtained during study phase I, and written consent was obtained during study phase II.

## 3. Results

### 3.1. Demographic and Health Characteristics

Overall, 3432 adults were recruited across the 13 regions of Saudi Arabia, of whom 51.2% were women and the mean age was 37.7 ± 11.7 (range: 18–87) years. In total, 20.3% of the participants were current smokers, and 25.2% had used a dietary supplement in the past 7 days. More details about the demographic and health characteristics of the participants are presented in Table 1.

### 3.2. Prevalence of Micronutrient Deficiencies

#### 3.2.1. Vitamin Deficiencies

The most prevalent vitamin deficiency was that of vitamin D (25 [O.H.] D) (64.3%) at ≤20 ng/mL, followed by vitamin B2 (44.8%) and vitamin A (9.6%). The lowest vitamin deficiency reported was that of vitamin B1 (2.3%), followed by vitamin B6 (2.4%) and vitamin B9 (3.5%) deficiencies (Table 2).

##### Vitamin Deficiencies by Sex

Vitamin D deficiency prevalence was higher in men (68.2%) than in women (59.2%), similar to vitamin B2 deficiency (Table 2). Conversely, vitamin A deficiency was more common among women than men (10.7% and 4.8%, respectively).

##### Vitamin Deficiencies by Age Group

Those in the 18–24- and 25–34-years age groups exhibited the highest prevalence of vitamin D deficiency (79.4% and 67.5%, respectively; Figure 1). Age differences were also observed in vitamin B1 deficiency, with the highest reported prevalence among those in the 35–44-years age group (4.6%), followed by the 25–34-years age group (3.0%), while the lowest prevalence was observed in the ≥65-years age group (1.3%). This difference was significant (*p* < 0.048). Age differences in the prevalence and patterns of the other vitamins were nonsignificant (Appendix A
Table A2).

#### 3.2.2. Mineral Deficiencies

In terms of the nutritional status of minerals, iron deficiency was the most prevalent (23.2%), followed by zinc (15.3%) and copper (8.7%) deficiencies (Table 3). Most participants exhibited normal magnesium levels (0.96%).

##### Mineral Deficiencies by Sex

Iron deficiency prevalence was significantly higher in women (39.2%) than in men (11.3%) (*p* < 0.001). Similar sex differences were observed for zinc deficiency, which was higher in women than in men (14.5% vs. 8.9%, *p* < 0.001). However, copper deficiency was more common in men than in women (12.6% vs. 4.9%; *p* < 0.001) (Table 4).

##### Mineral Deficiencies by Age Group

Among the age groups, the 18–24- and 35–44-years age groups exhibited the highest prevalence of iron deficiency (31.9% and 27.1%, respectively; Figure 2), significantly higher compared with that of other groups (*p* < 0.001). Age differences were also observed in magnesium deficiency, with the highest reported prevalence among those in the 55–64-years age group (2.4%), followed by the ≥65-years age group (1.3%), while the lowest prevalence was observed in the 45–54-years age group (0.2%). This difference was significant (*p* < 0.008). Age differences in the prevalence of all the other mineral deficiencies were nonsignificant (Appendix A
Table A2).

### 3.3. Heavy Metals

The measured levels of arsenic (99.7%) and mercury (99.9%) for most Saudi adults were within the normal ranges (Table 4).

#### 3.3.1. Heavy Metals by Sex

Men more commonly exhibited abnormal *arsenic* levels (0.3%) than women (0.1%); however, the difference was not statistically significant (*p* = 380). Similarly, abnormal mercury levels tended to be more common among men (0.4%) than among women (0.2%); however, the difference was not statistically significant (*p* = 0.476).

#### 3.3.2. Heavy Metals by Age Group

The 55–64 age group exhibited the highest prevalence of abnormal arsenic levels (0.4%), followed by the 35–44 (0.3%) age group; however, the difference was not statistically significant (*p* = 0.653). The highest prevalence of abnormal mercury levels was exhibited by the 18–24 age group (0.6%), followed by the 35–44 age group (0.5%); however, the difference was not statistically significant (*p* = 0.426). Appendix A
Table A3 includes details regarding the prevalence of abnormal arsenic and mercury levels based on age group.

## 4. Discussion

To the best of our knowledge, this is the first study to comprehensively evaluate the nutritional status of Saudi adults on a national level, particularly focusing on quantifying the micronutrient levels. Our results revealed that micronutrient deficiencies constitute a public health concern for Saudi adults, as at least 10% of the adults participating in the study exhibited four micronutrient deficiencies (vitamin D, vitamin B2, iron, and zinc). These results are largely consistent with earlier findings by the WHO Regional Office for the Eastern Mediterranean population [20] and those observed at the global level [6]. However, the arsenic and mercury levels observed in our population were not concerning.

Several vitamin shortfalls were evident, with deficiencies in vitamins D and B2 being the most prevalent. Although the definition of vitamin D deficiency is controversial [21], we applied the most common cutoff of vitamin D deficiency (25[O.H.]D ≤ 20 ng/mL) [21], which showed that the prevalence of vitamin D deficiency in the present study affected two-thirds of the study population. This is largely consistent with the results of a recent meta-analysis involving 16 studies conducted among 20,787 Saudi adults [22]. On comparing the prevalence of this deficiency in Saudi adults with the populations of neighboring countries, a lower prevalence was observed in Saudi Arabia than in Qatar (71.4%) [23], Oman (87.5%) [24], and Kuwait (83%) [25]. Internationally, however, the prevalence of this deficiency was higher in Saudi Arabia than in the United States (US) (41.6%) [26] and Europe (40.4%) [27]. The prevalence of vitamin D deficiency varied according to the age and sex of the participants, with younger adults and men exhibiting a higher vitamin D deficiency prevalence than older participants and women, which is consistent with the results found in the literature [28]. In contrast, our findings contradicted the only study conducted at the national level in Saudi Arabia, which revealed that older adults and women exhibited a higher prevalence of vitamin D deficiency [16]. This contradiction might be due to heterogeneity between both studies regarding age-predefined inclusion criteria and the vitamin cutoff used.

Extensive evidence highlighted that limited sunlight exposure is a chief contributor to vitamin D deficiency globally [29], as well as in Saudi Arabia [16,22,30]. Notably, several factors have been highlighted by minimal sun exposure, including the production of vitamin D due to skin pigmentation and minimal outdoor sun exposure due to the hot climate [16,31].

Studies on vitamin B2 deficiency are relatively limited [32], making comparison difficult owing to several factors, such as variation in age, population groups, or biochemical measurements [32,33,34,35]. Overall, our results indicate that 44.9% of Saudi adults exhibit vitamin B2 deficiency, which is lower than that reported in the United Kingdom (UK) (69%) [32] but higher than the estimated prevalence in the Korean population (21%) [34]. Generally, vitamin B2 deficiency is common in developing countries owing to the inadequate intake of riboflavin-rich products, such as milk and meat [32]. Other risk factors include various health conditions and metabolic disorders, smoking, pregnancy, lactation, and using hormonal contraceptives [32,36]. Vitamin B2 deficiency symptoms are nonspecific and vary from mild to severe, including skin disorders, hyperemia, mouth and throat edema, hair loss, migraine headache, heart failure, hypertension, brain dysfunction, and microcytic anemia [32,37].

Our findings indicate that approximately one-third of Saudi adults exhibit iron deficiency (23.2%), consistent with that reported in the literature [38], but higher than the global estimates [4,38]. Hwalla et al. [15] reported that in the Middle East the second highest prevalence of anemia in women of childbearing age was found in Saudi Arabia (40%), followed by Egypt (47.2%). In our findings, iron deficiency significantly differed based on age and sex, with women exhibiting a higher prevalence of iron deficiency than men (41.2% vs. 10.9%, respectively). This result was observed nationally [38], regionally [15], and internationally [4]. Furthermore, age differences exist in iron deficiency prevalence, particularly demonstrating a higher prevalence in young than in older adults; this is consistent with the results of other studies [4,15,39,40,41]. Risk factors associated with iron deficiency include the inadequate intake of iron-rich foods, blood loss (primarily menstrual or gastrointestinal), pregnancy, various health diseases, metabolic disorders, and older age [42]. Iron deficiency increases the risk of impaired immunity and anemia [42].

Zinc deficiency is widespread worldwide [4], especially in developing countries, where its prevalence reaches up to 30% [43]. The national estimate of zinc deficiency in Saudi Arabia is 15.3%, which is slightly lower than the estimated prevalence worldwide (≤17.3%) [43]. Zinc deficiency was characterized by sex differences, with women exhibiting a higher prevalence than men. These observations are consistent with Saudi and international studies [44,45]. Zinc deficiency has several risk factors, such as zinc loss or inadequate absorption and various chronic health conditions [46,47], and can engender growth retardation, respiratory infection, and impaired immune function [4].

Our study revealed no concerns regarding heavy metal levels (arsenic and mercury) among the Saudi Arabian adult population. Overall, regulatory agencies, such as the SFDA, implement monitoring programs to assess the levels of heavy metals, pesticides, and contaminants as well as their potential exposure and risk in manufactured goods and food [48]. A recent study involving food samples from Saudi markets reported that heavy metal content was within the permissible limits set by SFDA standards [11]. In addition, the SFDA regulates cosmetics, ensuring that cosmetic products manufactured in and imported to Saudi Arabia are free from heavy metals [49]. Thus, the low heavy metal levels observed in this study might be attributed to the implementation of such public health policies.

The high prevalence of several micronutrient deficiencies warrants urgent and effective public health interventions, including making healthy food more affordable and accessible and food fortification [4]. Fruits and vegetables are rich sources of numerous vitamins and minerals [50]. There is overwhelming evidence regarding the importance of food affordability in shaping food consumption behavior [51,52]. Multiple evidence-based practices are available to make food more affordable [53,54], several of which are suitable for this context. One study reported that only a small percentage of Saudi adults meet the dietary recommendations, especially for the intake of fruits (5.2%), vegetables (7.5%), and dairy products (26.2%) [55]. Fruits and vegetables can be made more accessible by reducing custom duty rates and subsidizing their price [56]. Other interventions could include food fortification [4,15,57,58], which is a safe and cost-effective strategy for improving diets and managing micronutrient deficiencies [4,15]. National strategies regarding micronutrient fortification are warranted to prevent micronutrient deficiency in Saudi Arabia and achieve a substantial public health benefit across the population. Unfortunately, a systematic review of food fortification implementation highlighted its ineffectiveness in managing micronutrient deficiency in most Middle Eastern countries [41]. Furthermore, a comparative study regarding the implementation of food fortification between Saudi Arabia and the US highlighted that current food fortification implementation methods in Saudi Arabia need to be improved [59].

A strength of this study is the fact that it involved a national sample, and, thus, the results are generalizable to the adult Saudi Arabian population Another strength is that this is the first study in Saudi Arabia aimed at determining vitamin and mineral deficiency prevalence and heavy metal levels among adults at a national level. However, it has some limitations. For example, study recruitment was suspended for over 9 months owing to the COVID-19 lockdown, which might have affected the dietary behavior and micronutrient status of the participants. Second, the study population primarily included individuals from urban cities across the 13 administrative regions of Saudi Arabia because of the availability of laboratories. Social desirability and recall bias could have affected the responses. However, the present data were chiefly based on laboratory findings. Finally, although the target sample size was 3536, only a sample size of 3432 was reached. This reflects the many challenges of achieving the target sample size, including difficulties in recruiting participants in some border regions. Generally, the actual sample size is close to the target sample size, at 3.0% lower than the target.

## 5. Conclusions

Our research yielded relevant, important, and timely findings regarding micronutrient deficiency status in Saudi adults at the national level. Several micronutrient deficiencies were identified. The most frequent ones include those of vitamins (vitamin B2 and D) and minerals (iron and zinc), which affect at least 1 in 10 Saudi adults. Strategies to prevent deficiencies are imperative and should be a high priority for health policymakers to improve the health of the population.

## Figures and Tables

**Figure 1 healthcare-10-02415-f001:**
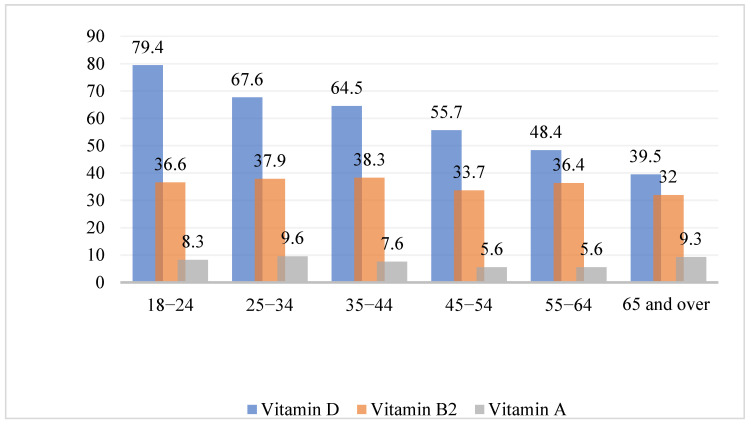
Distribution of vitamin D, B2, and A deficiencies based on age group.

**Figure 2 healthcare-10-02415-f002:**
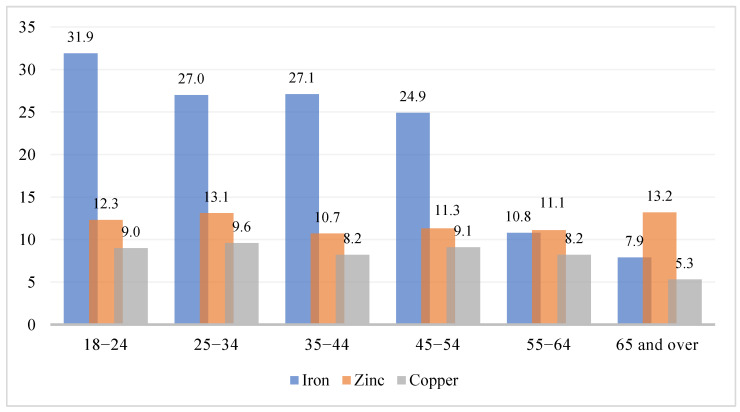
Prevalence of iron, zinc, and copper deficiencies based on age group.

**Table 1 healthcare-10-02415-t001:** Demographic and health characteristics of the study participants.

Characteristic	Men	Women	Total	Total (Weighted%)
	n (%)	n (%)	n (%)	
Sample size	1674 (48.8)	1758 (51.2)	3432 (100)	100
Age group				
18–24	130 (7.8)	224 (12.7)	354 (10.3)	13.9
25–34	579 (34.6)	547 (31.1)	1126 (21.8)	30.1
35-44	555 (33.2)	481 (27.4)	1036 (30.2)	29.0
45–54	259 (15.5)	332 (18.9)	591 (17.2)	17.0
55–64	118 (7.0)	131 (7.5)	249 (7.3)	7.9
≥65	33 (2.0)	43 (2.4)	76 (2.2)	2.1
Smoking and tobacco use status				
Never smoked	897 (56.8)	1646 (95.4)	2543 (76.9)	75.3
Previous smoker	151 (9.6)	11 (0.6)	162 (4.9)	4.4
Current smoker	532 (318)	68 (3.9)	600 (18.2)	20.3
Dietary supplements				
Had taken supplements within the past 7 days	341 (20.4)	495 (28.7)	836 (25.3)	25.2
History of chronic condition diagnoses				
Diabetes	155 (9.3)	143 (8.1)	298 (9.0)	8.5
Heart disease	24 (1.4)	15 (0.9)	39 (1.1)	1.2
Respiratory disease	48 (2.9)	60 (3.4)	108 (3.1)	3.6
Depression	20 (1.4)	32 (18)	52 (1.6)	2.9
History of morbidity				
Morbidity (≥1 chronic condition)	327 (20.7)	341 (19.8)	668 (20.2)	22.1
Multimorbidity (≥2 chronic conditions)	127 (7.6)	197 (11.2)	324 (9.8)	11.7

**Table 2 healthcare-10-02415-t002:** Prevalence of vitamin deficiencies among Saudi adults.

Vitamin Deficiency Variables	Men	Women	*p*-Value	Total	Total (Weighted%)
	n (%)	n (%)		n (%)	
Sample size	1674 (48.8)	1758 (51.2)		3432 (100)	100
A	81 (4.8)	188 (10.7)	<0.001	269 (7.9)	9.6
B1	54 (3.2)	57 (3.2)	0.974	111 (3.2)	2.3
B2	662 (39.5)	602 (34.2)	<0.001	1264 (36.9)	44.8
B6	20 (1.2)	37 (2.1)	0.57	57 (1.7)	2.4
B9	87 (5.2)	37 (2.1)	<0.001	124 (3.6)	3.5
B12	99 (5.9)	160 (9.1)	<0.001	259 (7.5)	8.1
D	1141 (68.2)	1041 (59.2)	<0.001	2182 (63.8)	64.3
E	50 (3.0)	57(3.3)	0.666	107 (3.1)	4.9

**Table 3 healthcare-10-02415-t003:** Prevalence of mineral deficiencies among Saudi adults.

Mineral Deficiency Variables	Men	Women	*p*-Value	Total	Total (Weighted%)
	n (%)	n (%)		n (%)	
Sample size	1674 (48.8)	1758 (51.2)		3432 (100)	100
Copper	211 (12.6)	86 (4.9)	<0.001	297 (8.8)	8.7
Iron	188 (11.3)	687 (39.2)	<0.001	875 (25.6)	23.2
Magnesium	10 (0.6)	12 (0.7)	0.753	22 (0.6)	0.4
Zinc	149 (8.9)	254 (14.4)	<0.001	403 (11.9)	15.3

**Table 4 healthcare-10-02415-t004:** Prevalence of arsenic and mercury levels among Saudi adults.

Heavy Metals	Classification	Men	Women	*p*-Value	Total	Total (Weighted%)
		n (%)	n (%)		n (%)	
Arsenic	Normal	1670 (99.8)	1756 (99.8)	0.380	3426 (99.8)	99.7
	High	4 (0.3)	2 (0.1)		6 (0.2)	0.3
Mercury	Normal	1664 (99.6)	1752 (97.8)	0.476	3416 (99.7)	99.9
	High	6.0 (0.4)	4 (0.2)		10 (0.3)	0.1

## Data Availability

Not applicable.

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
