# Peer review of "The Prevalence of Vitamin and Mineral Deficiencies and High Levels of Non-Essential Heavy Metals in Saudi Arabian Adults"

_healthcare, 2022, doi:10.3390/healthcare10122415_

Round 1

Reviewer 1 Report

Dear Authors,

The paper is well-written in terms of English. It is also well-structured. Nevertheless, some mistakes were not avoided:

Major points:

First of all, In my opinion the title and a lot of information contained in the text are poorly formulated. Below I present one of the many explanations of the term heavy metals  [e.g. Raychaudhuri, Sarmistha & Pramanick, Paulami & Talukder, Pratik & Basak, Apaala. (2021). Polyamines, Metallothioneins and Phytochelatins - Natural Defence of plants to mitigate heavy metals. 10.1016/B978-0-12-819487-4.00006-9].

“Heavy metals—Definition

Heavy metals (HMs) are defined as those elements having an atomic number greater than 20 and atomic density above 5 g cm− 3 and must exhibit the properties of metal [2,38,39]. The HMs can be broadly classified into two categories: essential and nonessential heavy metals. Essential HMs are those required by living organisms for carrying out the fundamental processes like growth, metabolism, and development of different organs. There are numerous essential heavy metals like Cu, Fe, Mn, Co, Zn, and Ni required by plants as they form cofactors that are structurally and functionally vital for enzymes and other proteins [40]. Essential elements are often required in trace amounts in the level of 10–15 ppm and are known as micronutrients [3]. Nonessential heavy metals like Cd, Pb, Hg, Cr, and Al are not required by plants, even in trace amounts, for any of the metabolic processes.”  

-The authors gave too general an analytical procedure for the determination of the elements and vitamins tested. Can it be expanded?

Minor points:

- Figure 1 and 2- you do not need a table with values (you can read them from the charts)

- Table 4 and Appendix - units are missing

- Line 173- the percent sign is missing

- References list – please adjust the list to the requirements of the journal and add where it is possible doi number without hyperlinks

After all corrections have been made, the manuscript should be published.

I take the time to thank you very much again for the opportunity of reviewing this paper.

Wish you all the best.

Sincerely,

The reviewer

Author Response

Thank you for the opportunity to submit a revised draft of our manuscript. We highly appreciate the time and effort that you and the reviewers have dedicated to providing your valuable feedback, and we are grateful for the insightful comments on our work. We have incorporated changes to reflect most of the suggestions provided and have highlighted the changes within the manuscript. Here is a point-by-point response to the reviewers’ comments and concerns.

Comment 1:

 First of all, In my opinion the title and a lot of information contained in the text are poorly formulated. Below I present one of the many explanations of the term heavy metals [e.g. Raychaudhuri, Sarmistha & Pramanick, Paulami & Talukder, Pratik & Basak, Apaala. (2021). Polyamines, Metallothioneins and Phytochelatins - Natural Defence of plants to mitigate heavy metals. 10.1016/B978-0-12-819487-4.00006-9].

“Heavy metals—DefinitionHeavy metals (HMs) are defined as those elements having an atomic number greater than 20 and atomic density above 5 g cm− 3 and must exhibit the properties of metal [2,38,39]. The HMs can be broadly classified into two categories: essential and nonessential heavy metals. Essential HMs are those required by living organisms for carrying out the fundamental processes like growth, metabolism, and development of different organs. There are numerous essential heavy metals like Cu, Fe, Mn, Co, Zn, and Ni required by plants as they form cofactors that are structurally and functionally vital for enzymes and other proteins [40]. Essential elements are often required in trace amounts in the level of 10–15 ppm and are known as micronutrients [3]. Nonessential heavy metals like Cd, Pb, Hg, Cr, and Al are not required by plants, even in trace amounts, for any of the metabolic processes.”

Response 1:

Regarding your comment on the title, we would welcome a suggested title but nevertheless revised it and hope it is satisfactory: ‘The prevalence of vitamin and mineral deficiencies and high levels of non-essential heavy metals in Saudi Arabian adults’.

Regarding the term heavy metals: At the beginning of our study, we clearly stated that we aimed to examine several vitamins (A, B1, B2, B6, B9, B12, D and E) and minerals (copper, zinc, magnesium and iron). Furthermore, we assessed the levels of two well-known toxic heavy metals: arsenic and mercury.

We believe that we were clear when we classified non-essential mineral elements (‘arsenic and mercury are well-known toxic nonessential heavy metals’, line 41). Such types of mineral elements are known as non-essential mineral elements. We know that some of the other minerals that we studied (namely, zinc and copper) are heavy metals; however, they are also classified as minerals, and these minerals are essential for organisms’ overall development. This is contrasted with mercury and arsenic for the sake of clear communication.

Below, we present a reference for reporting copper and zinc as classified as minerals in the health context.

https://link.springer.com/referenceworkentry/10.1007/978-3-030-30192-7_25

Below, we present a reference for how often mercury and arsenic are considered toxic heavy metals in the literature.

https://www.frontiersin.org/articles/10.3389/fphar.2021.643972/full

We believe that our title adjustment makes this information clear.

Comment 2:

 The authors gave too general an analytical procedure for the determination of the elements and vitamins tested. Can it be expanded?

Response 2:

Thank you for your comment. We believe the information about the determination of the elements and vitamins tested in our study is detailed in the paragraph on methods (lines 117–125), and we provided additional information by listing each laboratory test conducted in Appendix A (methodology, instrument, measured compound and ref. range). We do not know what else could be added to the information provided.

Comment 3

 [Figure 1 and 2- you do not need a table with values (you can read them from the charts)]

Response:

Thank you for this insightful comment. We agree and have modified the figure. We hope that it has been revised satisfactorily.

Comment 4:

 [- Table 4 and Appendix - units are missing

Response 4:

 Thank you for your comment. It has been revised accordingly.

Comment 5:

 [Line 173- the percent sign is missing

Response 5:

 Thank you for this observation. It has been revised accordingly (Line 177).

Comment 6:

[References list – please adjust the list to the requirements of the journal and add where it is possible doi number without hyperlinks

Response 6:

Thank you for this suggestion. The references have been formatted per ACS style, as required by the journal. The doi numbers without hyperlinks have been revised accordingly.

Reviewer 2 Report

This is the first study in Saudi Arabia aimed at determining vitamin and mineral deficiency prevalence and heavy metal levels among adults at a national level. This is a very important study. However, there are several concerns as follows.

1.     The authors estimated that a minimum sample size of 3536 participants would be sufficient to complete the study. However, this study included 3432 adults. Is this sufficient? Has Covid-19 affected the study enroll?

2.     Ref#16 found an overall vitamin D deficiency among 62.65% and 40.6% of female and male aged 15 years and older, respectively. In this study, vitamin D deficiency prevalence was higher in men (68.4%) than in women 157 (59.6%), which is consistent with the results found in the literature Ref#28. More explanation is needed in the discussion section.

3.     In Fig1, vitamin A or vitamin B12? Which one is correct?

4.     The authors mentioned that the 18–24 and 25–34 years age groups exhibited the highest prevalence of iron deficiency (32.7% and 26.3%, respectively). Actually, the 35–44 years age groups exhibited 27.1%.

5.     The authors mentioned that the 55–64 years age group exhibited the highest prevalence of abnormal arsenic levels (0.4%). However, isn’t it 0.2% in Appendix C?

6.     In Table 1, total number of 45–54 and 55–64 years age groups are 597 and 251, respectively. However, normal arsenic people numbers are 1036 in 45-54 age group and 590 in 55-64 in Appendix C, respectively. Please check.

7.     I think “3.3.2 Heavy metals by age group” and “Appendix C” need confirmation and correction.

Author Response

Dear Reviewer,

Thank you for the opportunity to submit a revised draft of our manuscript. We highly appreciate the time and effort that you and the reviewers have dedicated to providing your valuable feedback, and we are grateful for the insightful comments on our work. We have incorporated changes to reflect most of the suggestions provided and have highlighted the changes within the manuscript. Here is a point-by-point response to the reviewers’ comments and concerns.

Comment 1: The authors estimated that a minimum sample size of 3536 participants would be sufficient to complete the study. However, this study included 3432 adults. Is this sufficient? Has Covid-19 affected the study enroll?

Response 1: We appreciate this valuable comment. We have added this to our discussion section, lines 196–202, as one of the limitations of our study.

Lines 196-202: Finally, although the target sample size was 3,536, only a sample size of 3,432 was reached. This reflects the many challenges of achieving the target sample size, including difficulties recruiting participants in some border regions. Generally, the actual sample size is close to the target sample size, at 3.0% lower than the target.

Comment 2. Ref#16 found an overall vitamin D deficiency among 62.65% and 40.6% of female and male aged 15 years and older, respectively. In this study, vitamin D deficiency prevalence was higher in men (68.4%) than in women 157 (59.6%), which is consistent with the results found in the literature Ref#28. More explanation is needed in the discussion section.

Response 2: Thank you for this suggestion. It would have been interesting to explore this aspect. However, it seems slightly out of the scope of our study because Ref#16, which was a national study, is not consistent with our study regarding inclusion criteria in terms of its age category (≥ 15 years of age) versus our study (≥ 18 years of age). In addition, Ref#16 used a cut-off of ≤ 28 ng/mL for vitamin D (25[O.H.]D), and we applied the most common cut-off of vitamin D deficiency (25[O.H.]D ≤20 ng/mL). However, we agree with your suggestion and have incorporated it into the manuscript (lines 232–236).

Lines 232-236: In contrast, our findings contradicted the only study conducted at the national level in Saudi Arabia, which revealed that older adults and women exhibited a higher prevalence of vitamin D deficiency [16]. This contradiction might be due to heterogeneity between both studies regarding age-predefined inclusion criteria and the vitamin cut-off used.

Comment 3: In Fig1, vitamin A or vitamin B12? Which one is correct?

Response 3: Thank you for this insight. It is vitamin A, and we have corrected it accordingly.

Comment 4: The authors mentioned that the 18–24 and 25–34 years age groups exhibited the highest prevalence of iron deficiency (32.7% and 26.3%, respectively). Actually, the 35–44 years age groups exhibited 27.1%.

Response 4: Thank you for your comment. It has now been revised accordingly.

Comment 5: The authors mentioned that the 55–64 years age group exhibited the highest prevalence of abnormal arsenic levels (0.4%). However, isn’t it 0.2% in Appendix C?

Response 5: Thank you for your comment. It has now been revised accordingly.

Comment 6: In Table 1, total number of 45–54 and 55–64 years age groups are 597 and 251, respectively. However, normal arsenic people numbers are 1036 in 45-54 age group and 590 in 55-64 in Appendix C, respectively. Please check.

Response 6: Thank you for your comment. It has now been revised accordingly.

Comment 7: I think “3.3.2 Heavy metals by age group” and “Appendix C” need confirmation and correction.

Response 7: We thank the reviewer for this important point. We reported the findings regarding both age and gender by assigning an adjustment weight to each survey respondent (equal weight). However, we have now reported all the unweighted data in terms of age, gender and the total and hope that it is clear.

Sincerely,

Round 2

Reviewer 1 Report

Thank you for making the corrections.I have no more comments.

In this version, the paper is acceptable.